

# What has been missed for predicting human attention in viewing driving clips?

Jiawei Xu[1], Shigang Yue[1], Federica Menchinelli[2] and Kun Guo[2]

[1] School of Computer Science, University of Lincoln, Lincoln, United Kingdom
[2] School of Psychology, University of Lincoln, Lincoln, United Kingdom

## ABSTRACT

Recent research progress on the topic of human visual attention allocation in scene perception and its simulation is based mainly on studies with static images. However, natural vision requires us to extract visual information that constantly changes due to egocentric movements or dynamics of the world. It is unclear to what extent spatio-temporal regularity, an inherent regularity in dynamic vision, affects human gaze distribution and saliency computation in visual attention models. In this free-viewing eye-tracking study we manipulated the spatio-temporal regularity of traffic videos by presenting them in normal video sequence, reversed video sequence, normal frame sequence, and randomised frame sequence. The recorded human gaze allocation was then used as the 'ground truth' to examine the predictive ability of a number of state-of-the-art visual attention models. The analysis revealed high inter-observer agreement across individual human observers, but all the tested attention models performed significantly worse than humans. The inferior predictability of the models was evident from indistinguishable gaze prediction irrespective of stimuli presentation sequence, and weak central fixation bias. Our findings suggest that a realistic visual attention model for the processing of dynamic scenes should incorporate human visual sensitivity with spatio-temporal regularity and central fixation bias.

## INTRODUCTION

The amount of available visual information in our surroundings is often beyond our brain's processing capability. To effectively interact with our natural and social world, we selectively gaze at and process a limited number of local scene regions or visual items that are informative or interesting to us. As our gaze allocation is a sensitive index of attention, motivation and preference (*Henderson, 2007*), the fixated regions tend to have a distinct subjective perceptual quality which enables them to stand out from the neighbours, and the choice of these salient targets reflects our internal representation of the external world.

The central research question in this active visual exploration process is to understand how we choose the fixated regions in the scene. Psychological studies have suggested that both bottom-up local saliency computation and top-down cognitive processes are

Corresponding author
Kun Guo, kguo@lincoln.ac.uk

actively involved in determining our fixations in scene exploration (*Henderson, 2007*). The bottom-up signals include image immanent features that could transiently attract fixation irrespective of a particular task demand. For instance, we tend to avoid low-contrast and homogeneous 'predictable' regions in natural scenes, and bias our fixation to local features with high-contrast, high spatial frequency, high edge density, and complex local structure (e.g., curved lines, edges and corners, as well as occlusions or isolated spots) (*Mannan, Ruddock & Wooding, 1996*; *Reinagel & Zador, 1999*; *Parkhurst & Niebur, 2003*; *Acik et al., 2009*), or to local regions deviated from surrounding image statistics (*Einhäuser et al., 2006*). The top-down signals, such as an individual's expectation, experience, memory, semantic and task-related knowledge, are task-dependent and could significantly modulate our gaze allocation in scene exploration (e.g., *Guo et al., 2012*; *Pollux, Hall & Guo, 2014*; *Gavin, Houghton & Guo, 2017*). Recent studies have further proposed that fixation selection is uniquely driven by learned associations between stimuli and rewards (*Anderson, 2013*), and is influenced by aspects of innate human bias, such as the tendency to fixate human/animal face and body in the scene and to look more often at the central part of the scene (*Tatler et al., 2011*).

Motivated by these empirical studies, computer vision scientists have tried to develop human-vision-inspired foveated active artificial vision systems capable of working in real-time, and have proposed various computational models for predicting and/or mimicking human gaze allocation in natural vision (*Borji, Sihite & Itti, 2013*). Closely resembling our knowledge about neural processing in the early visual system, the widely cited bottom-up saliency model (*Itti & Koch, 2000*) compares local image intensity, colour and orientation through centre-surround filtering at eight spatial scales, combines them into a single salience (conspicuity) map with a winner-take-all network and inhibition-of-return, and then produces a sequence of predicted fixations that scan the scene in order of decreasing salience. To improve its relatively low level of predictive power (e.g., 57%–68% correct fixation prediction in some scene free-viewing tasks; *Betz et al., 2010*), aspects of innate human bias and top-down processing, such as face or object detection, scene context, natural statistics and learned associations between stimuli and rewards, are later incorporated into the model (e.g., *Torralba et al., 2006*; *Judd et al., 2009*; *Kanan et al., 2009*; *Goferman, Zelnik-Manor & Tal, 2012*). This combination of bottom-up saliency-driven information and top-down scene understanding has greatly improved gaze prediction in real-word visual search and free-viewing tasks.

However, the experimental findings discussed above and computational visual attention models in scene perception are derived mainly from studies using images in laboratory settings. In the real world we need to select, extract and process visual information that constantly changes due to egocentric movements or dynamics of the world. The motion cues are likely to attract visual attention and play a crucial role in subsequent scene understanding. Indeed, recent psychological studies have shown that motion and continuous temporal change are stronger predictors for gaze allocation than local image intensity or colour (*Carmi & Itti, 2006*; *Le Meur, Le Callet & Barba, 2007*; *Dorr et al., 2010*), and can enhance perceptual performance in understanding ambiguous visual cues, such as degraded or subtle facial expressions (*Cunningham & Wallraven, 2009*).

Currently, most of the visual attention models focus on spatial information processing with little consideration of temporal cues (*Borji, Sihite & Itti, 2014*), especially for the evolved spatial–temporal coherence over a period of time. For example, in the saliency-based visual attention system (*Itti, Koch & Niebur, 1998*), three feature channels—colour, intensity, and orientation, and multi-scale image features are combined into a single topographical saliency map, in order to break down the complex scene understanding problem by rapidly selecting conspicuous locations for detailed analysis. The attention based on information maximization (AIM) model uses Shannon's self-information measure to calculate the saliency of image regions (*Bruce & Tsotsos, 2006*), while the incremental coding length (ICL) approach measures the respective entropy gain of each feature to maximize the entropy of the sampled visual features in both dynamic and static scenes (*Hou & Zhang, 2008*). Recently a spatiotemporal saliency algorithm based on a centre-surround framework has been proposed in which the salience map was calculated using dynamic textures (*Mahadevan & Vasconcelos, 2010*). Marat's model (*Marat et al., 2009*) extracts two signals from the video stream corresponding to the two main outputs of the retina: parvocellular and magnocellular pathways, both signals are split into elementary feature maps by cortical-like filters. These maps are then fused into a spatio-temporal saliency map.

To further enhance the model's predictive power, we proposed a dynamic visual attention model (*Xu, Yue & Tang, 2013*) based on three channels. The first channel is motion intensity to reveal the faster moving objects. Then the spatial cues indicate the different moving objects in the spatial location, while the temporal cues denote the variability of one object in the temporal dimension. The rarity factor is imported to describe the unexpected or novel moving object that appeared in the visual field, which is likely to attract visual attention and should be captured or reflected in the saliency map, but to ignore the receding objects. The model has been developed further to analyze the attention shifts by fusing the top-down bias and bottom-up cues (*Xu & Yue, 2014*). The proposed model has five modules: the pre-learning process, top-down bias, bottom-up mechanism, multi-layer neural network and attention. The motion attention is estimated based on a motion vector field (MVF) comprising of three inductors: intensity inductor, spatial coherence inductor, and temporal coherence inductor (*Ma et al., 2005*). *Ban, Lee & Lee (2008)* also proposed a biologically motivated dynamic bottom-up selective attention model, which employs the maximum entropy algorithm to analyze the dynamics of the successive static saliency maps.

Although some of these models work reasonably well in simulated laboratory conditions, it is unclear how well they could predict human attention allocation in naturalistic conditions. For instance, these models tend to assign a high saliency rating to a moving object regardless of the predictability of motion trajectory, but we are unlikely to attend to a predictable movement (e.g., a travelling car with constant speed and predictable direction) for a prolonged period in natural environments. Given that our visual system is evolved to process dynamic spatiotemporal information, it is reasonable to assume that natural scene dynamics could elicit an optimal gaze strategy to facilitate scene understanding. Among various scene-dependent spatial and temporal cues in natural vision, spatio-temporal

regularity or predictability, in which objects around us often occur and move in statistically predictable ways to create a stream of visual inputs which are spatially and temporally coherent (e.g., the trajectory of a car moving on the motorway or an apple falling from a tree), is an inherent common regularity in the natural environment. Recent psychological and brain imaging studies have shown that the human visual system exploits this spatio-temporal regularity to effectively process dynamic visual inputs, such as detecting and discriminating the moving target (e.g., *Guo et al., 2004*; *Pollux & Guo, 2009*; *Hall, Pollux & Guo, 2010*; *Pollux et al., 2011*; *Roebuck, Bourke & Guo, 2014*).

To the best of our knowledge, we are not aware of any visual attention model that has examined the effect of movement predictability on its predictive power or has incorporated this spatio-temporal regularity into saliency computation. In this study, we installed a camera in the car and recorded typical driving scenes with different levels of traffic complexity. To systematically manipulate the videos' spatio-temporal regularity, each clip was presented in four different formats, such as normal video sequence, reversed video sequence, normal image sequence, and randomised image sequence. Such manipulation of the presentation sequence of the videos could selectively disrupt the temporal frame order (i.e., reversed video sequence), the spatial frame coherence (i.e., normal image sequence) or the spatio-temporal frame coherence (i.e., randomised image sequence) of the visual stream inputs. The gaze allocation of human observers viewing these clips was recorded and then used as the 'ground truth' to examine the predictive ability of the state-of-the-art visual attention models.

## METHODS

### Ethics statement

The Ethical Committee in School of Psychology, University of Lincoln, approved this study. Written informed consent was obtained from each participant prior to the study, and all procedures complied with the British Psychological Society Code of Ethics and Conduct. All the relevant measures, conditions and data exclusions have been reported, and the sample size was determined based on our previous research (*Röhrbein et al., 2015*).

### Eye-tracking study

Thirty-five undergraduate students (27 female, 8 male), age ranging from 18 to 28 years old with a mean of $19.4 \pm 0.37$ (Mean $\pm$ SEM), volunteered to participate in this study. All participants had normal or corrected-to-normal visual acuity, and normal colour vision (checked with Ishihara's Tests for Colour Deficiency, 24 Plates Edition).

Visual stimuli were presented on a non-interlaced gamma-corrected colour monitor (30 cd/m$^2$ background luminance, 100 Hz frame rate, Mitsubishi Diamond Pro 2070SB) with a resolution of $1,024 \times 768$ pixels. At a viewing distance of 57 cm, the monitor subtended a visual angle of $40 \times 30°$.

Thirty high-definition colour video clips were sampled from the authors' collection of traffic videos, taken by attaching a digital camcorder to the car windscreen. These video clips represent a variety of daily traffic conditions, such as approaching traffic lights, road junction and roundabout; low-speed driving at car park and residential area or on

inner-city road; high-speed driving on single carriageway, dual carriageway and motorway (see Fig. 1 for example frames from typical video scenes). Each video clip lasted 3 s (30 frames per second) and all clips were identical in size (720 × 404 pixels).

To examine to what extent the spatio-temporal regularity of the videos would affect our gaze allocation, each video clip was manipulated and later presented in four conditions: (1) *normal video sequence*: the clip was presented in its original, predictable sequence (from 1st frame to 90th frame); (2) *reversed video sequence*: the same clip was presented in reverse motion or time-reversed sequence (from 90th frame to 1st frame); (3) *normal image sequence*: six still shots were sampled at the beginning, middle and end of the video sequence (i.e., 1st, 16th, 31st, 46th, 61st and 76th frame), and then presented in their original order, each still image was displayed for 500 ms; (4) *randomised image sequence*: the same six still images were presented in a randomised sequence, each lasting 500 ms. As a result, 120 video clips were generated for the testing session (30 original clips × 4 manipulations). These clips were displayed once in a random order during the testing.

A free-viewing task was used to mimic natural viewing conditions. During the experiment the participants sat in a chair with their head restrained by a chin-rest, and viewed the display binocularly. To calibrate eye movement signals, a small red fixation point (FP, 0.3° diameter, 15 cd/m$^2$ luminance) was displayed randomly at one of 9 positions (3 × 3 matrix) across the monitor. The distance between adjacent FP positions was 10°. The participant was instructed to follow the FP and maintain fixation for 1 s. After the calibration procedure, the trial was started with four FPs presented at the corners of the screen area in which the videos were displayed. To minimize initial central fixation bias in scene viewing (*Tatler, 2007*), the participant was instructed to randomly choose one FP to fixate. If the participant maintained fixation for 1 s, the FP disappeared and a testing video clip was presented at the centre of the screen. During the free-viewing presentation, the participant was instructed to "view the video clip as you would normally do". No reinforcement was given during this procedure. All participants have viewed all of the 120 video clips.

Horizontal and vertical eye positions from the self-reported dominant eye (determined through the Hole-in-Card test or the Dolman method if necessary) were measured using a Video Eyetracker Toolbox (a camera-based system tracking pupil centre and corneal reflection) with 50 Hz sampling frequency and up to 0.25° accuracy (Cambridge Research Systems, UK; http://crsltd.com/tools-for-vision-science/eye-tracking/high-speed-video-eye-tracker-toolbox/). The software developed in Matlab computed horizontal and vertical eye displacement signals as a function of time to determine eye velocity and position (*Guo et al., 2006*). The location of gaze point within each video frame was extracted from the raw data.

## Data analysis and model comparison

Human gaze allocation in viewing of the normal video sequences was treated as the 'ground truth' or 'bench mark' to compare the participants' gaze behaviour on the four presentation conditions and the predictive power of the chosen computational models. The bench mark gaze location at each frame was averaged from 35 participants after

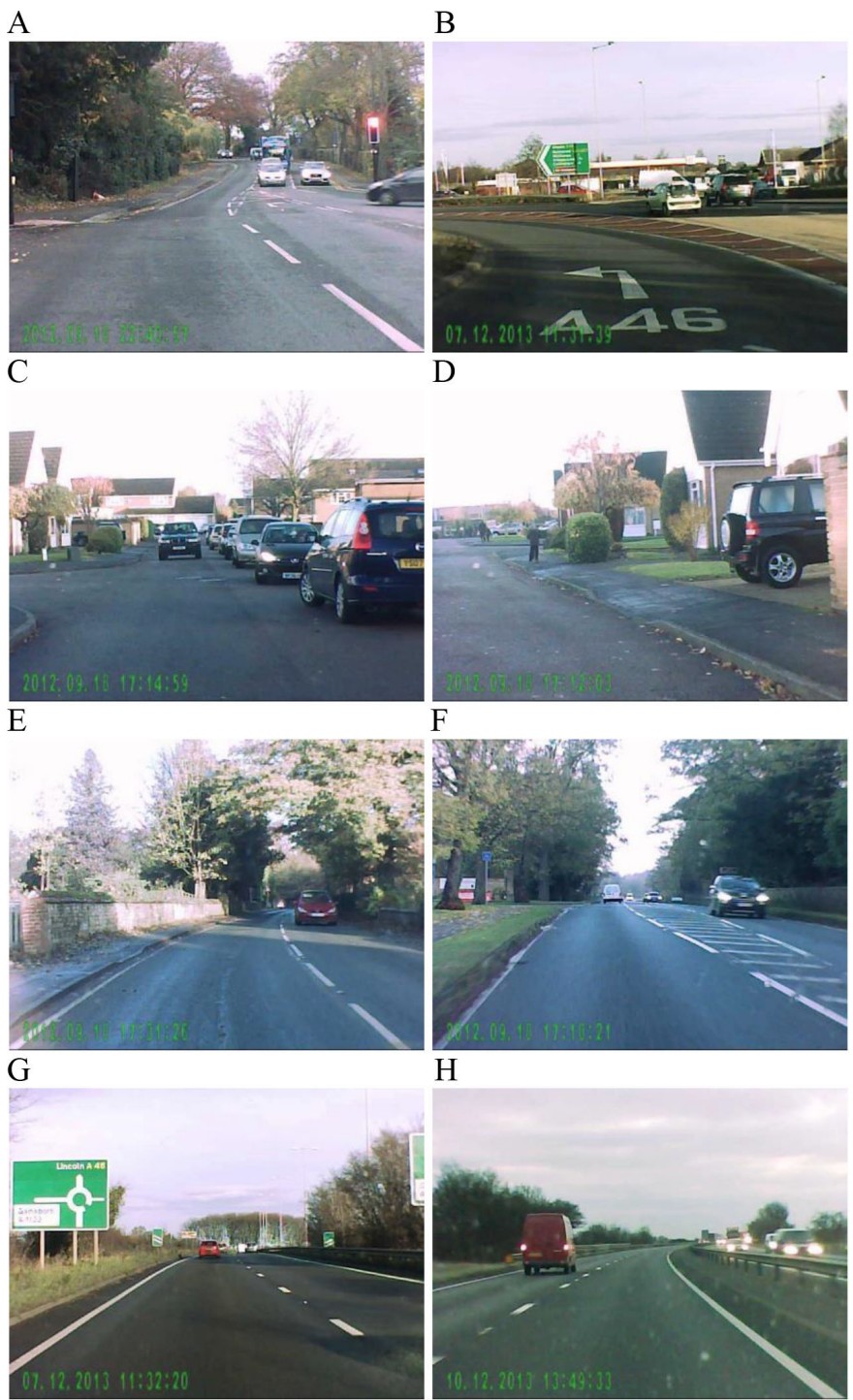

**Figure 1 Example frames from traffic videos used in this study.** The videos included clips representing approaching traffic lights (A) and a road junction (B), low-speed driving at a car park (C) and residential area (D), driving in an inner-city road (E and F), and high-speed driving in a dual carriageway (G) and motorway (H).

checking inter-observer agreement (or individual differences in gaze allocation) through the "all-except–one observers" protocol (i.e., how well the fixation distribution of each participant can be predicted by that of other participants; *Röhrbein et al., 2015*). We then adopted a frame-based similarity measurement to quantify the differences in comparison. For each frame within a given video sequence, we first calculated the Euclidean Distance ($D_f$) between bench mark gaze location ($x_0, y_0$) and predicted gaze location ($x_1, y_1$) from a chosen computational model.

$$D_f = \sqrt{(x_1 - x_0)^2 + (y_1 - y_0)^2}.$$

Then for the same video frame, we used the same equation to calculate the Euclidean Distance between bench mark gaze location ($x_0, y_0$) and actual gaze location ($x_1, y_1$) from each participant ($\check{D}_i$), and computed the averaged Euclidean Distance across all the participants ($\check{D}$).

$$\check{D} = \frac{1}{35} \left( \sum_{i=1,2,\ldots,35} \check{D}_i \right).$$

The similarity score ($S$) for this video frame was then computed as

$$S = \frac{\check{D}}{\check{D} + D_f}.$$

After repeating the same protocol for all the frames within this given video sequence, the final similarity score for the clip was then averaged across all the similarity measures for each frame in the video, and was ranged from 0 to 1, in which 0 represents no similarity at all and 1 means identical spatial–temporal gaze distribution between human observers and model prediction. This normalized approach for similarity measurement could minimize the impact of stimulus size or content on the similarity calculation, and facilitate data comparison across different stimulus conditions, different participant groups or different studies.

Normalized Scanpath Saliency (NSS) and Area under Curve (AUC) or Receiver Operating Characteristics (ROC) are other commonly used measures in the literature to compare the predictive power of visual attention models (*Röhrbein et al., 2015*). However, NSS is the average of the response values of human gaze allocation without considering spatial phase (*Rothenstein & Tsotsos, 2006*), and the computation of AUC or ROC is more suitable for static image rather than dynamic video analysis (e.g., unable to capture frame-by-frame dynamic changes in a model's predictive performance; *Green & Swets, 1966*; *Zou, O'Malley & Mauri, 2007*). In this study we therefore focused on comparing the similarity score calculated from different presentation conditions with different computational models. As discussed in the Introduction, very few visual attention models have incorporated motion cues in the saliency computation. Based on their significance in the literature, we chose to compare the predicitive power of six models from *Itti, Koch & Niebur (1998)*, *Bruce & Tsotsos (2006)*, *Hou & Zhang (2008)*, *Mahadevan & Vasconcelos (2010)*, *Marat et al. (2009)* and *Xu, Yue & Tang (2013)*. These models have all been constructed for dynamic displays and contain a 'motion' channel. For each tested

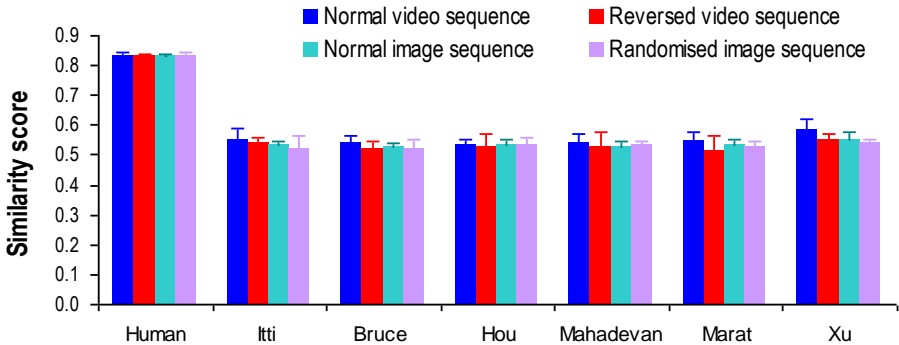

**Figure 2 Similarity scores from human observers and different visual attention models in computing saliency map of traffic videos and images with different presentation sequences.** Error bars represent standard deviation.

video clip, its saliency map (or predicted fixation allocation) was computed by these models separately using Matlab programmes with default settings obtained from the model developers.

## RESULTS

We first calculated the baseline of human performance using the similarity score, which measures how well the fixation distribution of each participant can be predicted by that of other participants (all-except-one observers). Specifically, we selected one participant's fixation allocations as actual fixations that we wanted to predict, and compared them with the averaged fixation allocations from the remaining participants. By repeating this procedure for all participants and averaging the resulting similarity scores, we obtained a measure for the variability (or inter-observer agreement) within all human gaze patterns that can serve as an upper bound to the performance of a given computational model. As shown in Fig. 2, irrespective of the stimuli presentation sequence, the gaze patterns from individual human participants were highly similar, with a similarity score distributed around 0.83, suggesting small human individual differences and a reliable 'ground truth' baseline or benchmark for model comparisons. The chance or random baseline was calculated by measuring the similarity score between actual human gaze allocation and randomly selected 'gaze' allocation at each frame for each video clip and presentation sequence. After repeating the same procedure 100 times, the averaged similarity score varied between 0.46 and 0.49 across different presentation sequences which was significantly below human baseline.

When compared with the averaged human gaze pattern and chance baseline, the predictive power of the tested models was clearly below the human baseline but was above the chance baseline regardless of presentation sequence (Fig. 2). To systematically compare the models' performance under different stimuli presentation sequences, a 7 (models: human baseline plus six visual attention models) ×4 (presentation sequences) repeated-measures analysis of variance (ANOVA) was conducted with similarity score as the dependent variable. Greenhouse–Geisser correction was applied where sphericity was

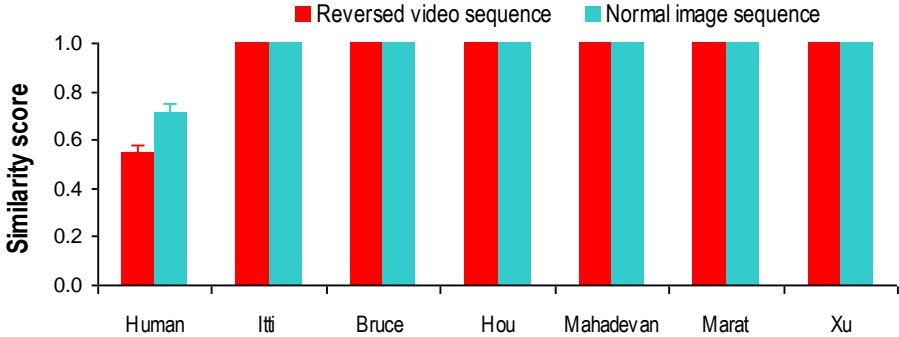

**Figure 3  Changes in similarity score over different stimulus presentation sequences.** For each human observer and each visual attention model, we computed the similarity score between fixation allocations (actual fixation allocations from human observers, and predicted fixation allocations from the models) in the normal video sequence and in the reversed video sequence or in the normal image sequence. Error bars represent standard deviation.

violated. The analysis revealed a significant main effect of model ($F(6, 174) = 1614.88$, $p < 0.001$, $\eta_p^2 = 0.98$) and presentation sequence ($F(2.03, 58.77) = 10.67$, $p < 0.001$, $\eta_p^2 = 0.27$), and a significant interaction between model and presentation sequence ($F(6.23, 180.67) = 4.2$, $p < 0.001$, $\eta_p^2 = 0.13$). Detailed post-hoc comparisons with Bonferroni correction further showed that all the visual attention models performed significantly worse than human baseline (all $ps < 0.01$). Among the tested models, the Xu model (*Xu, Yue & Tang, 2013*) performed slightly better than other models (all $ps < 0.01$), and the performance from the rest of models was indistinguishable (all $ps > 0.05$). Regarding stimulus presentation sequence, the normal video sequence induced better model performance for the *Bruce & Tsotsos (2006)*, *Marat et al. (2009)*, and *Xu, Yue & Tang (2013)* models (all $ps < 0.01$). All the models demonstrated the same predictive performance for the presentation of reversed video sequence, normal image sequence and randomised image sequence (all $ps > 0.01$, Fig. 2).

As clearly shown in Fig. 2, the tested computational visual attention models performed poorly across different video presentation sequences when using actual human gaze allocation as the baseline (or bench mark). To understand the potential causes for the models' inferior performance, we directly examined the impact of presentation sequence manipulation on actual and predicted gaze allocation by using gaze allocation in normal video sequence as the baseline. Specifically, for each human observer and each visual attention model, we computed the similarity score between fixation allocation (actual fixation allocations for human observers, and predicted fixation allocations for the models) in the normal video sequence and in the reversed video sequence, and between fixation allocation in the normal video sequence and in the normal image sequence (Fig. 3). Such analysis could directly inform the extent to which the disruption of temporal frame order (i.e., reversed video sequence) or spatial frame coherence (i.e., normal image sequence) of the visual stream inputs would affect actual and predicted gaze allocation.

A 7 (models) × 2 (presentation sequences) ANOVA revealed a significant main effect of model ($F(6, 174) = 7139.36$, $p < 0.001$, $\eta_p^2 = 0.99$) and presentation sequence

$(F(1, 29) = 273.28, p < 0.001, \eta_p^2 = 0.9)$, and a significant interaction between model and presentation sequence $(F(6, 174) = 273.29, p < 0.001, \eta_p^2 = 0.9)$. Specifically, manipulating presentation sequence clearly decreased similarity between actual human fixation distributions in the normal video sequence and in the reversed video sequence or in the normal image sequence, and such effect was more evident in the reversed video than in the normal image sequence (all $ps < 0.01$, Fig. 3), suggesting that temporal frame order might play a more important role in determining human gaze allocation in video-watching than spatial frame coherence. All the visual attention models, on the other hand, have predicted the same location of salient region for each video frame regardless of presentation sequence (i.e., indistinguishable high similarity scores when comparing between normal video sequence and reversed video sequence, or between normal video sequence and normal image sequence), suggesting that the crucial role of temporal frame order has been ignored in these computational models.

When exploring static images, humans tend to look more frequently at the central part of the image, especially at the initial stage of image viewing (*Tatler, 2007*; *Röhrbein et al., 2015*). This central fixation bias is mainly due to the image centre providing a convenient starting point to ensure rapid access to every point of interest in the image (*Tatler, 2007*). As demonstrated in Fig. 4, human observers seemed to show similar central location bias in video watching. Irrespective of video content, the spatial location of their gaze point was less varied across each frame and was more often close to the frame image centre in comparison with the predicted gaze location from the tested visual attention models. To quantify the difference between the spread of the fixations made from humans and models, for each video, we averaged both actual and predicted fixation distance from each frame's image centre (Fig. 5). The one-way ANOVA revealed a significant difference between the actual and predicted fixation distributions $(F(6, 174) = 3.35, p = 0.004, \eta_p^2 = 0.11)$. Specifically, humans demonstrated a stronger central fixation bias than the visual attention models (all $ps < 0.05$), but there was no clear difference in the predicted fixation distance from the image centre across the six tested models (all $ps > 0.05$). Clearly, this property of human central fixation bias could (at least) partly account for the inferior predictive performance of current computational models.

## DISCUSSION

In this eye-tracking study, we selectively disrupted the spatio-temporal regularity of typical traffic videos, and compared actual human gaze allocation with predicted gaze allocation by state-of-the-art visual attention models. Our analysis clearly demonstrated a high human inter-observer agreement for viewing the original and manipulated video clips (Fig. 2), suggesting a very similar attention allocation strategy across individuals in driving conditions. The tested computational visual attention models, however, performed poorly in comparison with human gaze behaviour. Specifically, the models showed indistinguishable gaze prediction irrespective of stimuli presentation sequence (Fig. 3), suggesting that the spatio-temporal regularity has not been appropriately implemented by these models. Furthermore, human gaze allocation demonstrated a clear central fixation

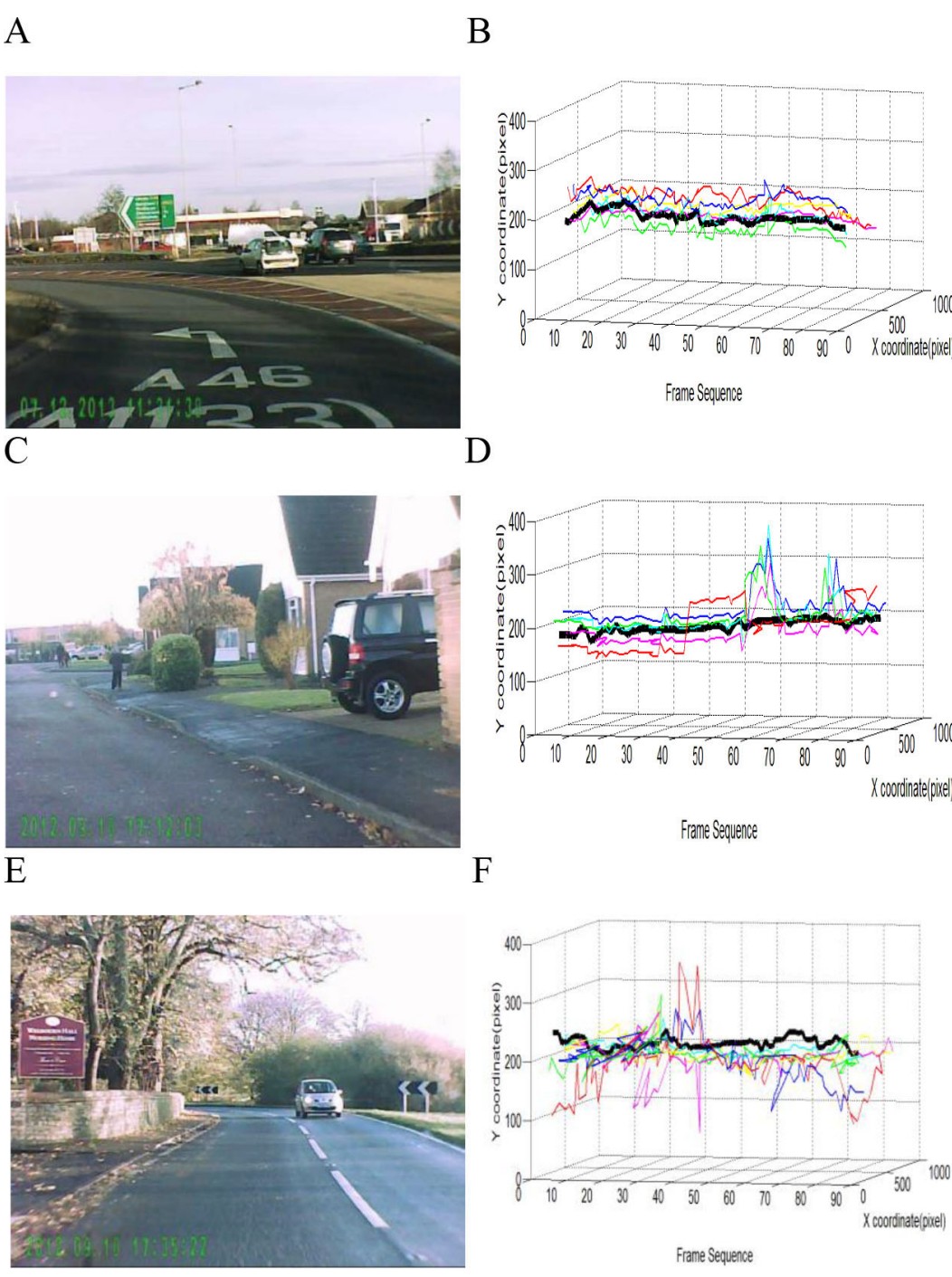

**Figure 4** **Distribution of actual and predicted gaze allocation for each frame of three typical videos (A and B, C and D, E and F) presented in normal video sequence.** The pictures are example frames extracted from videos representing junction approach (A), low-speed (C) and high-speed driving (E). The black lines in (B), (D), and (F) represent the averaged co-ordinates of human gaze allocation for each of the 90 frames. The coloured curves represent predicted gaze allocation from the six tested visual attention models.

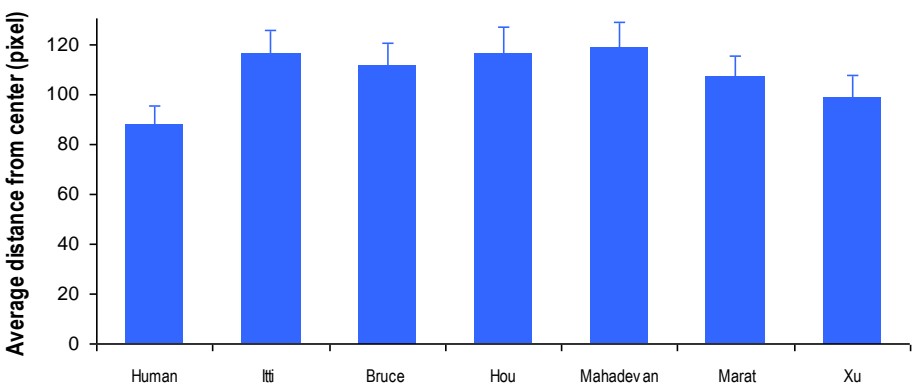

**Figure 5   Comparison of actual and predicted fixation distribution in viewing of traffic videos, using metrics of averaged fixation distance from the video frame centre.** Error bars represent standard error of the mean.

bias that was much stronger than in the gaze distribution predicted by the models (Fig. 5). It seems that the poor performance of visual attention models in processing of dynamic scenes was at least due to the lack of consideration of spatio-temporal regularity and central fixation bias.

As an inherent property of dynamic natural scenes, spatio-temporal regularity is routinely exploited by the human visual system to effectively process visual inputs. For instance, in driving situations we frequently expect that a particular object (e.g., a car in the motorway) will be presented at a particular location and time because of the spatial and temporal structure of the current scene, and prior knowledge of the visual world regularities (*Guo et al., 2004*). Consequently, we can easily direct our attention to the expected object and/or location to perform visual tracking or target analysis with increased perceptual sensitivity (*Cunningham & Wallraven, 2009*; *Guo et al., 2004*; *Hall, Pollux & Guo, 2010*), and ignore those objects that have high physical saliency but are not part of spatio-temporal regularity in dynamic scenes. For example, while watching driving videos or actually driving a car, experienced drivers allocate more fixations to the front and centre view than to the dashboard, right side or left side view (*Nabatilan et al., 2012*). They also tend to make predictive fixations by looking far ahead and visually targeting objects or locations on the desired trajectory or future path (*Kandil, Rotter & Lappe, 2010*; *Lappi et al., 2013*).

In this study, we further observed that in comparison with normal video sequence, manipulating spatio-temporal regularity by selectively disrupting temporal (i.e., reversed video sequence) or spatial frame order (i.e., normal image sequence) significantly altered human observers' attention allocation (Fig. 3). This effect was more evident in the presentation of reversed video sequences, suggesting a high sensitivity of the human visual attention system to the temporal sequence of predictable dynamic scenes. Such sensitivity could optimize visual processing efficiency and accuracy. Indeed, psychological studies have observed that the judgment of face gender, identity and expression in degraded or noisy presentation conditions is significantly impaired when reversing the facial movement video or altering its natural motion speed (*Lander & Bruce, 2000*; *Hill & Johnston, 2001*; *Cunningham & Wallraven, 2009*). Similarly, reversing the direction of movement impairs

object recognition in dynamic settings (*Vuong & Tarr, 2004*; *Wang & Zhang, 2010*). Clearly, spatio-temporal regularity plays a crucial role in attention allocation and associated scene perception in natural vision.

When exploring static scene images, human natural gaze behaviour often shows a central fixation bias, which can be ascribed to several factors. First, in photography, objects of interest are most often placed in the centre of the image. Second, the image centre as point of fixation could ensure rapid access to every point of interest in the image. It might therefore be advantageous to start scene viewing in the centre. The motor bias, the tendency to make shorter and more horizontal saccades, does not play a big role in central fixation bias (*Tatler, 2007*). This central fixation bias is often incorporated in the visual attention models for processing static scenes (*Borji, Sihite & Itti, 2014*). When viewing dynamic driving videos, our observers also demonstrated a similar 'central' fixation bias. They tended to allocate attention to the horizontal plane close to eye level, and paid little attention to the model-predicted salient region above or below the eye level. This finding is in agreement with recent studies demonstrating a tendency to maintain gaze near the centre of the display when watching a wide range of video clips, such as natural or urban scenes, sports, cartoon and film clips (*Le Meur, Le Callet & Barba, 2007*; *Berg et al., 2009*; *Tseng et al., 2009*; *Dorr et al., 2010*). It seems that central fixation bias is a spontaneous human gaze behavior that is independent of image/video content and cognitive task.

Although using computationally different methods to compute local saliency, all the tested visual attention models performed far below the human baseline, suggesting that they cannot realistically predict human attention allocation in viewing of (at least) dynamic driving clips. Furthermore, for each video frame these models predicted the same local saliency region or object irrespective of stimuli presentation sequence (i.e., normal video sequence vs reversed video sequence vs normal image sequence), indicating that the video's spatio-temporal regularity has not been appropriately implemented in the computational models. Considering that the human visual system applies spatial–temporal regularity as part of top-down guidance for determining gaze allocation, it seems that this factor, like other top-down factors such as knowledge of scene context and image statistics distribution (*Torralba et al., 2006*; *Judd et al., 2009*; *Kanan et al., 2009*; *Goferman, Zelnik-Manor & Tal, 2012*), should also be incorporated in a realistic manner into the visual attention model to cope with predictable dynamic visual inputs, possibly through visual motion sensitive neuron models (e.g., directional selective visual neuron models; *Rind, 1990*) and/or depth sensitive neuron models (*Rind & Bramwell, 1996*; *Gabbiani & Krapp, 2006*; *Yue & Rind, 2006*; *Yue & Rind, 2013*). It should be noted, however, that the driving video clips only represent part of the visual motion signals in our natural environment. The nature of human driving behaviour may create a strong central fixation bias and subsequently amplify its role in visual saliency computation. The global optic flow associated with driving clips may also disrupt those attention models designed to detect local featural motion. The extent to which we can generalize current findings to other types of motion signals remains to be seen.

### Funding

This work was supported by the grants of EU FP7-IRSES Project LIVCODE (295151) and HAZCEPT (318907), and Horizon 2020 projects ENRICHME (643691) and STEP2DYNA (691154). The funders had no role in study design, data collection and analysis, decision to publish, or preparation of the manuscript.

### Grant Disclosures

The following grant information was disclosed by the authors:
EU FP7-IRSES Project LIVCODE: 295151.
EU FP7-IRSES Project HAZCEPT: 318907.
Horizon 2020 project ENRICHME: 643691.
Horizon 2020 project STEP2DYNA: 691154.

### Competing Interests

The authors declare there are no competing interests.

### Author Contributions

- Jiawei Xu performed the experiments, analyzed the data, contributed reagents/materials/analysis tools, wrote the paper, prepared figures and/or tables.
- Shigang Yue conceived and designed the experiments, contributed reagents/materials/analysis tools, wrote the paper, prepared figures and/or tables, reviewed drafts of the paper.
- Federica Menchinelli performed the experiments.
- Kun Guo conceived and designed the experiments, analyzed the data, contributed reagents/materials/analysis tools, wrote the paper, prepared figures and/or tables, reviewed drafts of the paper.

### Human Ethics

The following information was supplied relating to ethical approvals (i.e., approving body and any reference numbers):

The Ethical Committee in School of Psychology, University of Lincoln approved this study. Written informed consent was obtained from each participant prior to the study, and all procedures complied with the British Psychological Society Code of Ethics and Conduct.

### Data Availability

The raw data has been supplied as a Data S1.

### Supplemental Information

Supplemental information for this article can be found online at http://dx.doi.org/10.7717/peerj.2946#supplemental-information.

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
