# Peer review of "What has been missed for predicting human attention in viewing driving clips?"

_PeerJ, doi:10.7717/peerj.2946_

## Round 0.1 · original submission · Major Revisions

I have received thorough reviews from two experts in the field. Both have major concerns about the manuscript that you will need to address in a revision. The greatest concern, as I see it, is that you seem to be testing something of a straw-man by comparing models constructed for static images to those constructed for dynamic displays. You either need to address the motivation for including the static image models or drop them from your analyses. Regardless of which direction you choose, your descriptions of the models need to include far more detail, with a special focus on parameters related to the primary manipulation in your experiment.

I would like you to add a statement to the paper confirming whether you have reported all measures, conditions, data exclusions, and how you determined your sample size. You should, of course, add any additional text to ensure the statement is accurate. This is the standard reviewer disclosure request endorsed by the Center for Open Science [see http://osf.io/project/hadz3]. I include it in every review.

The reviews are quite clear and can speak for themselves. In addition to addressing the substantive items from the reviews, I hope you will also work to fix the grammatical errors throughout the paper.

·

Basic reporting

The submission adheres to all policies and is generally self-contained and suitable.

The English does need to be checked in places, and even in the title. "What have been missed..." should be changed to either "What has been missed..." or "What factors have been missed..." or similar. In fact, the title is not a clear reflection of the stimuli or research question.

Experimental design

The research question and the gap being filled needs to be explained in more detail. For example, from lines 112 to 132 the authors argue that we don't know whether models should incorporated "spatio-temporal coherence" to better predict fixation. This is a bit confusing because on the previous page the authors mention several saliency models which include motion features, as well as their own model which includes "coherence inductors". I think a clearer explanation is needed about what motion features have previously been modelled, whether these correlate with fixations, and what the authors are suggesting which is different.

There are a number of method details which are not described sufficiently.
1. Did all participants see all clips? Or just a selection?
2. What hardware was used for the eye tracking? Is the toolbox described just the software?

Validity of the findings

The main findings concern the similarity between human fixations and model predictions across many frames. I have concerns about how these calculations were derived. Although similarity score data are provided, the actual original eye position data are not, as far as I can tell. My concerns are as follows:

1. The authors measure the distance between two points and convert it into a similarity score (equations 1 and 2, line 201). The units of this are not specified, but regardless I don't see why it is scaled in this way. For example, if the D is 1 degree, equation 1 would give 1 / 2 = 0.5. If D = 2 (twice the distance), equation 1 gives 1 / 3 = 0.33 (more than half the similarity). It is not clear why this metric, which does not scale linearly, has been chosen. This is not a conventional approach, so more justification is necessary, perhaps with reference to work by people like Mannan, who tried to quantify such distances in the past.

2. Line 204 states that the "benchmark gaze location was averaged from 35 participants". This is problematic. Imagine that half the participants fixated at x=100 and half the participants fixated at x=600. The average (mean?) location is x=350, even though nobody actually fixated this location at all! So this analysis is either not explained properly or is incorrect.

3. Given these weaknesses, the authors either need to use a more standard approach, or they need to better explain why the AUC or other measures are not sufficient. On line 213 they argue that such measures are only suitable for static images, but given that the authors area analysing frame by frame anyway, this doesn't seem to be a problem.

4. Later on in the paper, the authors refer to "fixation maps". This does not seem to reflect the details in the "Date analysis" (should be "Data") subsection.

5. Is there a way to put a lower bound on the similarity score? Typically, previous research has used a random or chance baseline, which would tell us whether the models are performing above chance.

6. Figure 3 is confusing. Presumably what we want to know is whether the models are better for normal vs. reversed sequences, and line 252 suggests that it is. Is this not already shown in Figure 2?

7. The plots in Figure 4 do not really show the variation in x and y very clearly.

Reviewer 2 ·

Basic reporting

Overall style is fine.
The manuscript could use a copy-editing round to catch a few non-idiomatic phrasings, mostly involving the use of articles ("a", "the") such as in line 34 or lines 85/86.

Experimental design

The experimental design is OK in the sense that it is appropriate for assessing the comparisons that the authors have chosen to set up. I say considerably more in the “validity” section of this review, which is the focus of my concerns.

Validity of the findings

The authors seek to find empirical evidence for something that is intuitively obvious – that human eye movements are a function of more than just the distributions of certain “usual suspects” of features in static images.

That a proposition seems to be intuitively obvious is not a reason to fail to test that proposition, and the authors are to be commended for trying to bring some methodological rigor to the question of how eye movements vary with video sequences as opposed to static images.

The overall tenor of the paper, however, leaves me underwhelmed. As noted in the comments about experimental design, the issue is not that the authors are “wrong” in the sense of having made clear-cut mistakes. Rather, the combination of (1) stimulus selection (video sequences), stimulus manipulation (perturbations of base sequences), and (3) model selection makes the impact of the core theses of the manuscript weak.

Re: (1) stimulus selection – driving sequences with a camera mounted to point along the axis of instantaneous vehicle motion guarantees a central fixation bias. Moreover that bias is probably different from the central fixation bias that accompanies the vast bulk of “snapshots” that have been composed by human operators to emphasize a “subject” of the image that was subsequently posted online. A driving video also generates different types of motion, including global optic flow that is disrupted by motions of independently moving objects. But such categories of motion are not likely to be singled out by models that incorporate only measures of local featural motion.

Re: (2) stimulus manipulation – running video backwards creates certain anomalies that might be expected to affect selection of eye fixations in certain sequences, but it is not clear that this manipulation is the only (or best) probe for what the authors claim to be interested in. Indeed, nowhere in the manuscript do they clarify what they mean by “spatio-temporal coherence” and how that might be different from “motion”. Consideration of the gestalt property of common fate makes clear that not all arrays of motion are coherent. Similarly, the freezing of individual frames in some sort of sequence (normal or reverse) is an interesting manipulation, but it is not at all clear what aspects of which temporal events in an original image sequence might be sufficiently adequately sampled, and which ones not, to possibly affect selection of eye fixations.

Re: (3) here the manuscript is just bizarre, as three of the models chosen for testing do not include motion. Given the evidence that motion can affect eye movements, not having a “motion” input to a model would seem enough to disqualify it from consideration, just like one would not expect a model with only a single, monocular processing channel to do very well in a depth-from-stereo task where other models are getting two disparate images. With this said, the idea that the performance of the three models that do contain motion is only a little better (in one case) than the performance of the purely static models suggest that motion alone is not what makes human performance different. But now we have come full circle. The choice of driving video sequences seems pretty arbitrary, and it’s not clear which model would do what, when confronted with other sorts of video input.

Additional comments

I have just written the kind of review that I hate to receive. My comments on validity amount to a plea that you had written some other paper, with some other stimuli, and some other manipulations. It’s not that what you did is “wrong”, but rather that the results are so inconclusive and likely to be a function of the juxtaposition of very particular videos with some particular models.

I support and encourage the general investigation of the importance of spatiotemporal coherence in eye movements. There is certainly a vast literature in the “laboratory” study of eye movements concerning transitions from saccadic to smooth-pursuit eye movements that may be worth exploiting – but here again, that would be a very different study than you have done, as smooth pursuit eye movements do not seem to factor into the design of the present study.

Perhaps you could recast your choice of stimuli to focus on the difference between “mere” motion salience (i.e., the appearance of some transient burst of motion) vs. a more elaborate form of motion coherence that affords some predictive power for planning of eye movements.

I’m sorry that I cannot be more enthusiastic about the study as reported.

Small point:
Line 103 – please explain “the rarity factor”

---

## Round 0.2 · Minor Revisions

This manuscript is greatly improved from the previous version. There are a few lingering issues that the reviewer has identified. I, like the reviewer, am unclear on the rationale behind the scaling factor. Secondarily, I would like you to, once again, proofread your introductory content. There are lingering grammatical issues that make it hard to track. For example, in the final paragraph of page 5, you refer to "incremental codling length," when you probably mean "incremental coding length," and in the first paragraph of page 6, you say that "temporal cues donate the variability," when I think you mean "denote the variability."

·

Basic reporting

No further comments.

Experimental design

The authors have addressed my previous points.

RE: my point #2, I think it would be better to include a sentence, after the webpage for the eye tracker, which describes the hardware. Readers will want to know if is this is a camera based system, whether it tracks pupil and corneal reflection etc. They shouldn't have to go to a webpage to find this.

Validity of the findings

The authors have made some changes. However, I still found it very difficult to understand what the similarity measure is doing. They have changed the equation and state (line 220) that "D is the averaged Euclidean Distance between actual and predicted gaze locations across all the frames for a given video sequence, and df is the Euclidean Distance between actual (or bench mark) gaze location ( x0 , y0 ) and predicted gaze location ( x1 , y1 )". This makes it very hard to see why D and df are not just the same quantity. Surely what we want to know is whether predicted locations are close to actual gaze locations (in which case D will be small). What is the motivation for including a different scaling factor.

I would advise the authors to include a figure showing an example of this calculation for one frame. This would make it much more clear. The figure could also more clearly illustrate the different comparisons that are made.

---

## Round 0.3 · accepted · Accept

Thank you for continuing to improve this document across drafts. I believe that this final draft is much stronger than the first and will make a valuable contribution to the literature.